# Effectiveness of Surgical Treatment on Survival of Patients with Malignant Pleural Mesothelioma

**DOI:** 10.3390/cancers17142360

**Published:** 2025-07-16

**Authors:** Renata Báez-Saldaña, María Esther Marmolejo-Torres, Marco Antonio Iñiguez-García, Aída Jiménez-Corona, Juan Alberto Berrios-Mejía

**Affiliations:** 1Teaching Department, Instituto Nacional de Enfermedades Respiratorias, Ismael Cosío Villegas, Mexico City 14080, Mexico; dramarmolejom@gmail.com; 2División de Posgrado, Facultad de Medicina, Universidad Nacional Autónoma de México, Mexico City 04510, Mexico; markcardio@hotmail.com; 3Thoracic Surgery Department, Instituto Nacional de Enfermedades Respiratorias, Ismael Cosío Villegas, Mexico City 14080, Mexico; juanalbertoberrios@gmail.com; 4Departamento de Epidemiología Ocular y Salud Visual, Instituto de Oftalmología, Fundación Conde de Valenciana IAP, Mexico City 06800, Mexico; aidaajc@gmail.com; 5Dirección General de Epidemiología, Secretaría de Salud, Mexico City 01480, Mexico

**Keywords:** malignant pleural mesothelioma, surgery, pleurectomy/decortication, extrapleural pneumonectomy, survival

## Abstract

The role of surgery in treating malignant pleural mesothelioma (MPM) remains debated due to limited evidence from clinical trials. In this study, we examined long-term survival in patients with MPM who received surgery combined with neoadjuvant chemotherapy compared to those treated with chemotherapy alone. Of the 122 patients, only 16 underwent surgery; eight received pleurectomy/decortication (PD), and eight had extrapleural pneumonectomy (EPP). At five years, the survival rate was higher in the surgery group (53%) than in the chemotherapy-only group (23%). Patients treated with PD achieved the best outcomes, with a five-year survival rate of 67%, compared to 40% for EPP. After adjusting for relevant clinical factors, surgery was linked to a significantly lower risk of death. These findings suggest that surgical treatment, especially PD, can improve survival in selected patients with MPM.

## 1. Introduction

Malignant pleural mesothelioma (MPM) is a rare tumor derived from pleural mesothelial cells. There is strong evidence of its association with asbestos exposure, although there are increasing cases in individuals without a clear history of asbestos exposure. According to GLOBOCAN figures, in 2020, there were 30,870 incident cases and 26,278 deaths worldwide. That same year, the incidence of cases in Latin America and the Caribbean was 1238 cases (4% of total cases in the world) and 1082 deaths (4.1% of all cancer deaths in the world). In Mexico, an age-standardized incidence rate was estimated to be 0.20–0.79 cases per 100,000 population for men and 0.12–0.24 cases for women (GLOBOCAN, https://gco.iarc.fr/, accessed 12 October 2023) [1].

The median survival from diagnosis of MPM is 6 to 9 months [2]. Compared to supportive care, chemotherapy results in a 12-month survival rate versus a 9-month survival rate [3]. Surgery is an option for patients at an early stage of the disease and with good functional status. The goal of surgery is to achieve gross complete resection (GCR) of the tumor through either extrapleural pneumonectomy (EPP) or pleurectomy/decortication (PD). The choice between the two depends on the surgeon’s preferences rather than scientific evidence. The optimal surgical treatment for individual patients with mesothelioma has not yet been fully established, as existing scientific evidence is based on a limited number of clinical trials and small, highly varied case series with inconsistent results. One retrospective case series indicated that surgery was associated with longer overall survival [adjusted hazard ratio 0.64 (0.61–0.67) [4]. The Mesothelioma and Radical Surgery (MARS) clinical trial found no benefit in terms of survival or quality of life for patients treated with EPP as part of trimodality therapy compared to chemotherapy alone, with median survivals of 18 months and 19.5 months, respectively [5]. Additionally, a recent trial comparing outcomes in patients who underwent pleurectomy plus chemotherapy versus chemotherapy alone showed no difference in progression-free or overall survival, with 84.5% experiencing at least one surgical complication [6]. Studies comparing EPP and PD demonstrate similar survival results, with lower postoperative morbidity and mortality associated with PD [7,8,9]. According to a systematic review of patients treated with EPP, median survival ranged from 9.4 to 27.5 months, and 5-year survival rates ranged from 0% to 24% [10].

In our institution, surgery is performed on patients with mesothelioma who have good functional status, epithelioid histology, and locally advanced disease. However, there is no institutional report regarding this intervention or the survival of these cases compared to those who have received only chemotherapy. The objective was to examine the long-term survival of patients with malignant pleural mesothelioma who underwent surgical treatment plus chemotherapy versus chemotherapy alone.

## 2. Methods

This study is a secondary analysis of an existing dataset and a retrospective review of clinical records related to a biomarkers study in mesothelioma [11]. The original study was approved by the Research Committee (19 CI 012 013) and the Research Ethics Committee (CONBIOÉTICA-09-CEI-003-20160427) at the National Institute of Respiratory Diseases, Ismael Cosio Villegas, with approval code C30-12 dated 28 June 2012. Data collection occurred between 2017 and 2018. Due to the retrospective nature of the study, obtaining informed consent from participants was not feasible. However, the Ethics Committee granted a waiver of informed consent. The study adhered to the principles outlined in the Declaration of Helsinki 1975 and followed the guidelines of Good Clinical Practice.

The study included patients diagnosed with malignant pleural mesothelioma confirmed by histopathology between January 2012 and May 2015. This research was carried out at the National Institute of Respiratory Diseases in Mexico, a referral center primarily serving low-income patients from Mexico City and nearby states. Patients were referred by healthcare providers, physicians, or self-referred. All patients gave informed consent for hospitalization and for each medical procedure, whether surgical or non-surgical.

A total of 122 cases of malignant pleural mesothelioma were examined. Pathologists with over 20 years of experience in pulmonary pathology confirmed the diagnoses histopathologically. Samples were obtained through various procedures: thoracoscopy in 50 cases (41%), closed pleural biopsy in 35 cases (28.7%), ultrasound-guided biopsy in 24 cases (19.7%), thoracotomy in 8 cases (6.5%), and open biopsy in 5 cases (4.1%). All cases underwent chest imaging studies, including frontal chest X-rays and computed tomography scans of the chest and upper abdomen. For cases considered for surgical treatment, PET-CT studies were performed along with lymph node staging via mediastinoscopy or Endobronchial Ultrasound Bronchoscopy (EBUS). Additionally, all patients received a pre- and post-operative pulmonary rehabilitation program.

A multidisciplinary team at our institution collectively decided the treatment plan for each patient to diagnose and treat chest neoplasms. The team included pulmonologists, chest surgeons, oncologists, radiologists, and pathologists. Most patients received chemotherapy, and in select cases, surgery and radiotherapy. Some patients only received supportive care focused on symptom management because they chose not to undergo chemotherapy. Patients were considered for surgery if they had a good functional status, epithelioid histology, and a clinical stage of I, II, or IIIA according to the seventh edition of the TNM classification for malignant pleural mesothelioma. The type of surgery performed was either extrapleural pneumonectomy (EPP) or pleurectomy/decortication (PD); the choice depended on the surgeon’s preference. The same group of mesothelioma surgeons performed both EPP and PD procedures. In our study, pleurectomy/decortication (PD) was defined as the macroscopic complete removal of the parietal and visceral pleura while preserving the lung, following the recommendations of the International Mesothelioma Interest Group (IMIG) [12]. Extended PD (ePD), which involves resection of the diaphragm and/or pericardium, was not coded explicitly in our dataset.

### 2.1. Study Variables

Patients’ general characteristics (age, sex, comorbidities, history of smoking, wood smoke, and asbestos exposure at work and home) were collected using a standardized form. The following information was documented: symptoms and duration of the current condition, diagnostic procedures used to obtain the histological sample, histopathology results, functional status according to the Eastern Cooperative Oncology Group (ECOG) scale, clinical stage of the disease based on the TNM classification for mesothelioma, clinical laboratory studies (complete blood count and blood chemistry tests), initial antineoplastic treatment, surgical treatment, type of surgical procedure, surgical complications, and vital status.

The patient’s vital status was determined by the date of the last consultation. If the time since the last appointment was one month or more, based on the file review date, a phone call was made to the patient or her family.

The source of information was the relevant clinical record, in accordance with the Institute’s regulations and the standards of the Official Mexican Standard for the Clinical Record.

### 2.2. Statistical Analysis

Clinical and laboratory characteristics were compared at baseline between the study groups (surgery versus chemotherapy and supportive care). Comparisons for continuous variables were conducted using Student’s *t*-test (reporting means and standard deviations [SD]) or Wilcoxon rank-sum test (reporting medians and interquartile ranges [IQR], 25th and 75th percentiles) when appropriate. Pearson’s chi-square test was employed for categorical variables. Patients were followed from baseline until the outcome (death), loss to follow-up, or end of the study, with a median follow-up of 5 months (range 0.1 to 75.7). The mortality rate and 95% confidence interval (95% CI) were calculated. Kaplan–Meier survival analyses were performed to evaluate the association between surgery and time to death. The log-rank test was used to compare survival curves in patients with and without surgery. The Cox proportional hazards model was applied to all cases (*n* = 122), and a separate model was used for patients with clinical stages I, II, and IIIA (*n* = 62) to assess the relationship between surgery and mortality. Both models accounted for age, the duration from symptom onset to diagnosis (in days), anemia, the neutrophil-to-lymphocyte ratio, and ECOG scores greater than 2.

The proportional hazards assumption was assessed through the scaled Schoenfeld residuals, and the model’s overall fit was evaluated graphically using the cumulative risk function with Cox-Snell residues. All analyses used Stata 17 (Stata Corporation, College Station, TX, USA) and SAS 9.1 (SAS Institute Inc., Cary, NC, USA).

## 3. Results

A total of 122 cases with a mean age (SD) of 63 (±12) years were analyzed. The variables that showed significant differences between the surgery and chemotherapy groups included age (56 vs. 64 years), a lower frequency of any comorbidity (18.8% vs. 52.8%), and a reduced prevalence of systemic hypertension (6.3% vs. 31.1%). Similarly, the neutrophil/lymphocyte ratio was lower (2.3 vs. 4.3). The other clinical variables compared between the two groups did not show statistically significant differences (Table 1).

Among histopathology types, epithelioid was most common, with 117 (96%) cases, followed by sarcomatoid and mixed or biphasic, with 3 (2.4%) and 2 (1.6%) cases, respectively.

According to the TNM classification, there was a higher frequency of clinical stages IB, II, and IIIA in the group that underwent surgery compared to the chemotherapy group (Table 2).

Regarding treatment, 97 cases received chemotherapy, and 23 received supportive care, which involved symptom management. The chemotherapy regimens included platinum plus pemetrexed in 41 cases, and platinum plus gemcitabine or vinorelbine in 56 cases. Surgery was performed in 16 cases, with 8 undergoing EPP and 8 undergoing PD. Eight patients received radiotherapy; among them, 2 had PD, 5 had EPP, and 1 was from the chemotherapy group. Median survival for the surgery group was 24.2 months, compared to 7.7 months in the non-surgery group (Table 3).

Postoperative morbidity was comparable between the EPP and PD groups, with 5 out of 8 patients (62.5%) in each group experiencing postoperative complications. The median (IQR) length of hospital stay was 18 (16–21) days for patients who underwent PD, and 22 (15–26) days for those who underwent EPP. We did not observe significant differences in postoperative air leak duration or protein abnormalities between the two groups.

The median (interquartile range 25–75) of survival among those who underwent PD was 33 (16–49) months versus 14 (10–37) months in the EPP group, with a statistically significant difference (*p* = 0.0031).

Survival in patients who underwent surgery as a treatment for mesothelioma was significantly longer than in the chemotherapy group. At five years, survival for the former was 53% (95% CI 15–81%) vs. 23% (95% CI 10–40). The five-year survival among those with PD vs. EPP was 67% vs. 40%, respectively. The crude mortality rate at one year and two years was 0% and 17% vs. 17% and 17%, respectively (Table 4) (Figure 1 and Figure 2).

In the bivariate analysis, hemoglobin < 10 g (HR = 2.89, 95% CI 1.46–5.70, *p* = 0.002), lymphopenia (HR = 2.87, 95% CI 1.41–3.31, *p* < 0.001), neutrophil/lymphocyte ratio ≥ 6 (HR = 2.15, 95% CI 1.41–3.31, *p* = 0.001), albumin ≤ 3 g/dL (HR = 2.30, 95% CI 1.56–3.37, *p* < 0.001), and stage IV (HR = 2.0, 95% CI 1.39–2.92, *p* = 0.000) were positively associated with mortality. In contrast, surgery (HR = 0.40, 95% CI 0.23–0.70, *p* = 0.001), chemotherapy (HR = 0.24, 95% CI 0.15–0.38, *p* < 0.001), and radiotherapy (HR = 0.35, 95% CI 0.17–0.73, *p* = 0.005) were protective factors (Table 5).

In the multivariate analysis, undergoing either of the two types of surgical treatment was associated with a favorable survival outcome (HR = 0.34, 95% CI 0.19–0.61, *p* < 0.001) after adjusting for the following factors: age over 65, time from symptom onset to diagnosis, hemoglobin level below 10 g/dL, neutrophil-to-lymphocyte ratio ≥ 6, and ECOG score of 2 or higher. The benefits of surgery remain consistent and show a similar trend when analyzing only patients with stage I, II, and IIIA disease (Table 6).

This same trend was observed in a second multivariate analysis based on surgery type, using the same variables as the first model. The results showed an HR of 0.26, 95% CI 0.12–0.57, *p* = 0.001 for PD, and an HR of 0.48, 95% CI 0.22–1.06, *p* = 0.070 for EPP. Additionally, the findings for patients in clinical stages I, II, and IIIA align with the surgical benefit (Table 7).

## 4. Discussion

The role of surgery in malignant pleural mesothelioma (MPM) remains controversial due to the scarcity of high-quality randomized trials. In our study, surgical treatment—particularly pleurectomy/decortication (PD)—was associated with improved survival compared to chemotherapy alone. At five years, the survival rate was 53% (95% CI, 15–81%) for patients treated with surgery and neoadjuvant chemotherapy, versus 23% (95% CI, 10–40%) for those who received only chemotherapy. Among surgical procedures, PD showed a survival rate of 67%, compared to 40% for extrapleural pneumonectomy (EPP).

After adjusting for relevant covariates—age > 65, delay in diagnosis, hemoglobin < 10 g/dL, neutrophil-to-lymphocyte ratio > 6, and ECOG ≥ 2—surgery remained significantly associated with better survival (HR 0.34; 95% CI, 0.19–0.61). When analyzed separately, PD conferred a more pronounced benefit (HR 0.26; 95% CI, 0.12–0.57) than EPP (HR 0.48; 95% CI, 0.22–1.06).

The 2018 American Society of Clinical Oncology (ASCO) guidelines recommend surgery for selected patients with resectable MPM and good performance status, aiming for maximal cytoreduction, though without endorsing a specific surgical technique [7]. While it is established that surgery offers survival benefits compared to no treatment [4], evidence from clinical trials, especially regarding EPP, remains conflicting. For example, a randomized trial evaluating EPP in MPM reported an increased mortality rate (adjusted HR, 2.75; 95% CI, 1.21–6.26; *p* = 0.016) [5]. Consequently, EPP has been largely replaced by less radical approaches such as PD, which is associated with fewer perioperative complications and, as supported by our findings and previous studies [13,14], potentially improved outcomes.

PD consists of parietal and visceral pleurectomy without the goal of complete macroscopic resection, unlike EPP, which involves en bloc resection of the lung, pleura, diaphragm, and pericardium when necessary. Although observational studies have shown survival benefits associated with PD, the absence of definitive randomized evidence limits the generalizability of these findings. The recent MARS-2 multicenter trial, for instance, found no improvement in progression-free or overall survival with PD plus chemotherapy compared to chemotherapy alone, and reported a high complication rate (84.5%) [6]. However, interpretation of these findings must consider potential selection bias—14.2% of patients had non-epithelioid histology, 7.7% were N2, and 3.6% had M1 disease—which could have influenced the outcomes. In our cohort, both PD and EPP were associated with improved survival, with PD showing a significantly longer median survival (33 vs. 14 months, *p* = 0.0031). These results reinforce the relevance of PD as a potentially effective surgical approach in appropriately selected patients.

Trimodality therapy—induction chemotherapy followed by EPP and hemithoracic radiotherapy—has shown mixed results. While some observational studies report a median survival of 29 months in highly selected patients [15,16], a randomized trial found no benefit from adding radiotherapy to neoadjuvant chemotherapy and EPP, concluding that it added toxicity without survival improvement [17]. In our cohort, radiotherapy was administered to 7 of 16 patients who underwent surgery and to only one patient in the non-surgical group. Although radiotherapy appeared as a protective factor in the unadjusted analysis (HR = 0.35, 95% CI: 0.17–0.73; *p* = 0.005), this association was completely lost in the multivariable model (HR = 0.96, 95% CI: 0.36–2.57; *p* = 0.998), suggesting that the apparent benefit was likely confounded by other prognostic variables and patient selection.

Although our study did not include intraoperative therapies like hyperthermic intrathoracic chemotherapy (HITHOC), our findings are consistent with recent reports that advocate surgical intervention, especially PD, as a key part of multimodal treatment in MPM. At our center, PD was performed through open thoracotomy, since VATS-PD was not standard practice during the study period due to limitations in training and resources. While VATS approaches are increasingly preferred for reducing perioperative morbidity and length of stay, long-term oncologic outcomes remain uncertain and highly depend on the operator.

Emerging data suggest that minimally invasive surgery, combined with adjunctive therapies, may provide additional benefits. A randomized pilot study found improved survival with VATS-PD plus HITHOC compared to talc pleurodesis [18], and a systematic review supports increased survival and longer disease-free intervals with this combination [19]. VATS-PD has also been shown to be an independent predictor of better prognosis compared to biopsy alone [20]. While our study did not include these techniques, our findings support the survival advantage of surgery even without them. However, we recognize that future research should explore whether less invasive approaches, such as VATS and multimodal intraoperative strategies, can further improve patient outcomes while maintaining oncologic effectiveness.

These observations highlight the importance of personalized treatment strategies in MPM. Our study indicates that surgical intervention—particularly PD—may be a key predictor of survival. However, several limitations must be recognized. The retrospective, single-center design and the relatively small number of surgical cases limit how widely applicable our findings are. Data on surgical margins and quality of life were not available, preventing a more thorough evaluation of outcomes. Additionally, our dataset did not differentiate between standard PD and extended PD (ePD), which involves diaphragm and/or pericardial resection. Although all PD procedures aimed for complete macroscopic resection, we could not determine if ePD was performed in any cases, emphasizing the need for standardized operative documentation in future prospective studies.

Lastly, the observed survival advantage may have been influenced by early-stage disease and good functional status among surgical candidates. To mitigate this potential bias, we performed a stage-matched analysis with non-surgical patients and observed consistent results.

Despite these limitations, our findings are consistent with existing evidence and unlikely to be due to chance.

## 5. Conclusions

In conclusion, surgical treatment—especially PD—was linked to a significant survival benefit in selected MPM patients. After controlling for clinical and laboratory prognostic factors, surgery remained an independent predictor of improved survival (HR 0.34; 95% CI, 0.19–0.61). These findings highlight the importance of surgical intervention in managing MPM and support further research into less invasive techniques and multimodal strategies to enhance patient outcomes.

## Figures and Tables

**Figure 1 cancers-17-02360-f001:**
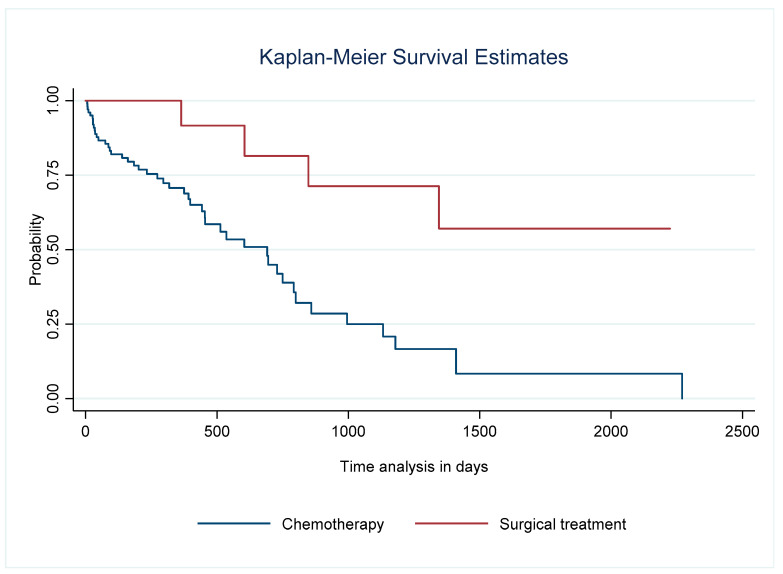
Graph of survival of patients with pleural mesothelioma with surgical treatment (decortication pleurectomy or extrapleural pneumonectomy) vs. chemotherapy. Log-rank test comparison, *p* = 0.0026.

**Figure 2 cancers-17-02360-f002:**
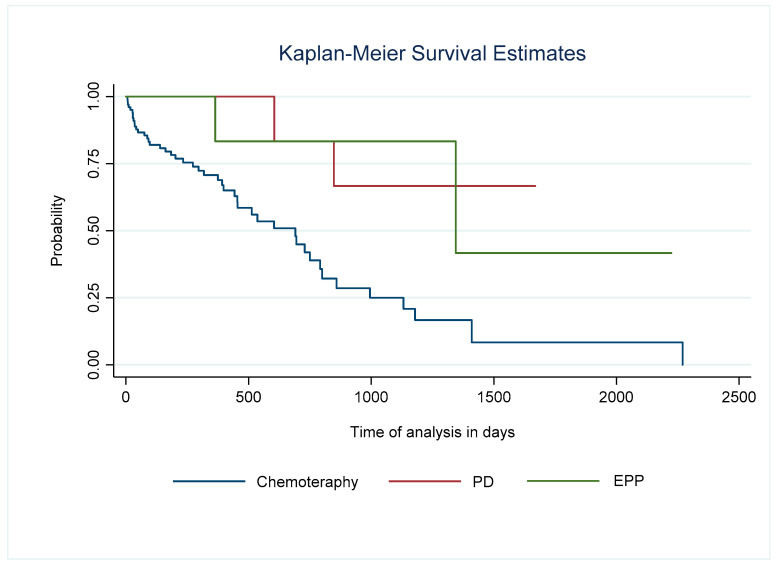
Graph of survival of patients who underwent pleurectomy/decortication and extrapleural pneumonectomy vs. chemotherapy. Log-rank test comparison, *p* = 0.0103 PD Pleurectomy/Decortication. EPP Extrapleural pneumonectomy.

**Table 1 cancers-17-02360-t001:** Clinical characteristics of mesothelioma patients.

Variables	Study Population *N* = 122	Surgery *n* = 16 (13.11%)	Chemotherapy or Supportive Care *n* = 106 (86.9%)	*p* Value ^a^
Age	63 (12)	56 (8)	64 (12)	0.019 ^b^
Sex, men *n* (%)	87 (71%)	10 (62.5%)	77 (72.6)	0.403
Any type of exposure to asbestos	85 (69.7%)	9 (56.3%)	76 (71.7%)	0.210
Current or past smoking *n* (%)	72 (59%)	8 (50%)	64 (60.4%)	0.431
Smoking index ^b^	7.5 (1.5–20)	12.7 (1.5–20)	7.2 (1.5–20)	0.295
Wood smoke *n* (%)	42 (34.4%)	4 (25%)	38 (35.9%)	0.395
Wood smoke index	42.5 (20–146)	20 (6.5–35)	54 (30–150)	0.051 ^c^
Comorbidities				
Any comorbidity	59 (48.4%)	3 (18.8%)	56 (52.8%)	0.011
Diabetes	26 (21.3%)	2 (12.5%)	24 (22.6%)	0.356
Hipertension	34 (27.9%)	1 (6.3%)	33 (31.1%)	0.039
Time from onset of symptoms to diagnosis (days) ^b^	149 (87–257)	175 (92–435.5)	125 (82–244)	0.216
Clinical laboratory studies				
Lymphopenia	21 (17.2%)	1 (6.3%)	20 (18.9%)	0.213
Neutrophil/lymphocyte ratio ^b^	3.9 (2.8–6.1)	2.3 (1.8–3.41)	4.3 (3.1–6.4)	0.001
Hemoglobin < 10 g/dL	10 (8.2%)	1 (6.3%)	9 (8.5%)	0.761
Albumin ≤ 3 g/dL	46 (37.7%)	4 (25%)	42 (39.6%)	0.261
ECOG ≥ 2	107 (87.7%)	14 (87.5%)	93 (87.7%)	0.979
Deaths	50 (41%)	4 (25%)	46 (43.4%)	0.163

Variables are in frequencies and percentages unless another form of summary measure is specified. ^a^ Chi-square test, ^b^ mean (standard deviation), Student’s *t* test, ^c^ median (interquartile interval, 25–75), Mann–Whitney U test.

**Table 2 cancers-17-02360-t002:** TNM classification and clinical staging.

Variables	Study Population *N* = 122	Surgery *n* = 16 (13.11%)	Chemotherapy or Supportive Care *n* = 106 (86.9%)	*p* Value ^a^
TNM classification				
T1	6 (4.9%)	2 (12.5%)	4 (3.8%)	
T2	35 (28.7%)	10 (62.5%)	25 (23.6%)	0.001
T3	41 (33.6%)	4 (25%)	37 (34.9%)	
T4	40 (32.8%)		40 (37.7%)	
N0	23 (18.9%)	8 (50%)	15 (14.2%)	
N1	20 (16.4%)	3 (18.8%)	17 (16%)	0.005
N2	72 (59%)	5 (31.2%)	67 (63.2%)	
N3	7 (5.7%)		7 (6.6%)	
M0	89 (73%)	16 (100%)	73 (68.9%)	
M1	27 (22.1%)		27 (25.5%)	0.033
Mx	6 (4.9%)		6 (5.6%)	
Clinical stage				
IB	2 (1.6%)	1 (6.2%)	1 (0.9%)	
II	13 (10.7%)	7 (43.8%)	6 (5.7%)	0.000
IIIA	47 (38.5%)	8 (50%)	39 (36.8%)	
IV	60 (49.2%)		60 (56.6%)	

Variables are in frequencies and percentages. ^a^ Chi-square test.

**Table 3 cancers-17-02360-t003:** Treatment modalities of cases with pleural mesothelioma according to surgery status.

	Total Population *N* = 122	Surgery *n* = 16 (16.4%)	Chemotherapy or Supportive Care *n* = 106 (83.6%)
Chemotherapy	97 (79.5%)	16 (100%)	83 (78.3%)
Pemetrexed-based chemotherapy	41 (42.3%)	8/16 (50%)	33/83 (39.8%)
Chemotherapy lines ^a^	2 (1–3) Min–max (1–6)	2 (1–4) Min–max (1–5)	2 (1–3) Min–max (1–6)
Chemotherapy complications ^b^	48/97 (49.5%)	5/14 (35.7%)	43/83 (51.8%)
Time in days from the start of oncological treatment to progression, median (IQR) ^a^	199 (112–336)	345 (99–592)	177 (112–291)
Radiotherapy	8 (6.6%)	7 (43.8%)	1 (0.9%)
Type of surgery			
Pleurectomy/Decortication		8 (50%)	
Extrapleural pneumonectomy		8 (50%)	
Time in months from diagnosis to death ^a^	9.5 (2.3–17.9)	24.2 (11.6–46.3)	7.7 (1.5–15.5)

Variables are in frequencies and percentages unless another form of summary measure is specified ^a^ Median (interquartile interval). ^b^ Anemia, neutropenia, and infection.

**Table 4 cancers-17-02360-t004:** Survival in cases of pleural mesothelioma in the total population and according to treatment status.

	1 Year	2 Years	3 Years	4–5 Years	6 Years
Survival (%) (95% CI)					
Total population *n* = 122	74 (65–82)	52 (40–63)	35 (22–48)	19 (8–34)	6 (0–32)
Surgery *n* = 16	100	82 (45–95)	71 (34–90)	53 (15–81)	53 (15–81)
Chemotherapy *n* = 97	79 (68–86)	59 (45–70)	41 (27–55)	23 (10–40)	8 (0–36)
Pleurectomy/ Decortication *n* = 8	100	83 (27–97)	67 (19–90)	67 (19–90)	–
Extrapleural pneumonectomy *n* = 8	100	80 (20–97)	80 (20–97)	40 (1–83)	40 (1–83)

**Table 5 cancers-17-02360-t005:** Unadjusted Cox proportional hazards of the association between mortality from malignant pleural mesothelioma and selected variables.

Variables	HR	95% CI	*p* Value
Age > 65	1.24	0.87–1.79	0.230
Sex	0.93	0.62–1.38	0.710
Any comorbitidy	1.21	0.84–1.73	0.304
Diabetes	1.20	0.78–1.87	0.404
Hipertension	1.04	0.69–1.55	0.860
Time in months from onset of symptoms to diagnosis	1.03	0.99–1.06	0.064
Hemoglobin < 10 g	2.89	1.46–5.70	0.002
Lymphopenia	2.87	1.41–3.31	<0.001
Neutrophil/lymphocyte ratio ≥ 6	2.15	1.41–3.31	0.000
Albumin ≤ 3 g/dL	2.30	1.56–3.37	<0.001
LDH in pleural fluid > 300 IU/L	2.33	1.45–3.75	0.000
pH in pleural fluid ≤ 7.2	1.47	0.89–2.42	0.136
Histological types			
Epithelioid	1		
Sarcomatoid	3.59	1.10–11.62	0.033
Mixed or biphasic	0.68	0.17–2.78	0.595
Clinical stage IV versus I, II and III	2.0	1.39–2.92	0.000
ECOG ≥ 2	1.53	0.89–2.65	0.123
Chemotherapy	0.24	0.15–0.38	<0.001
Radiotherapy	0.35	0.17–0.73	0.005
Surgical treatment (both PD and EPP)	0.40	0.23–0.70	0.001
Type of treatment			
Chemotherapy	1		
Pleurectomy/Decortication	0.35	0.17–0.74	0.006
Extrapleural pneumonectomy	0.48	0.17–74	0.048

**Table 6 cancers-17-02360-t006:** Multivariate Cox proportional hazards model of the association between both surgery methods and mortality from malignant pleural mesothelioma.

Variables	All Patients *n* = 122			Only Patients’ Clinical Stage I, II and III *n* = 62		
	HR	95% CI	*p*	HR	95% CI	*p*
Surgery (PD or EPP)	0.34	0.19–0.61	<0.001	0.37	0.19–0.72	0.003
Age > 65	0.92	0.63–1.35	0.682	0.73	0.38–1.38	0.332
Time in months from onset of symptoms to diagnosis	1.00	0.99–1.00	0.249	1.00	0.99–1.00	0.238
Hemoglobin < 10 g	3.90	1.91–7.96	<0.001	4.84	1.69–13.9	0.003
Neutrophil/lymphocyte ratio ≥ 6	2.22	1.44–3.42	<0.001	2.19	1.09–4.38	0.027
ECOG ≥ 2	1.70	0.97–2.96	0.062	1.47	0.72–3.01	0.295

PD Pleurectomy/Decortication. EPP Extrapleural pneumonectomy.

**Table 7 cancers-17-02360-t007:** Multivariate Cox proportional hazards model of the association between type of surgical treatment and mortality from malignant pleural mesothelioma.

Variables	All Patients *n* = 122			Only Patients Clinical Stage I, II and III *n* = 62		
	HR	95% CI	*p*	HR	95% CI	*p*
Type of treatment						
Chemotherapy	1			1		
Pleurectomy/Decortication	0.26	0.12–0.57	0.001	0.29	0.13–0.67	0.003
Extrapleural pneumonectomy	0.48	0.22–1.06	0.070	0.53	0.22–1.23	0.140
Age > 65	0.94	0.64–1.38	0.747	0.75	0.40–1.44	0.391
Time in months from onset of symptoms to diagnosis	1.00	0.99–1.00	0.296	1.00	0.99–1.00	0.302
Hemoglobin < 10 g/dL	4.12	2.0–8.46	<0.001	5.24	1.81–15.19	0.002
Neutrophil/lymphocyte ratio	2.31	1.49–3.59	<0.001	2.40	1.17–4.9	0.017
ECOG ≥ 2	1.71	0.98–2.98	0.060	1.47	0.72–3.01	0.294

## Data Availability

The data presented in this study are available upon reasonable request from the corresponding author and with permission from the Research Ethics Committee of the National Institute of Respiratory Diseases, Ismael Cosio Villegas. The data are not publicly available due to confidentiality policies related to ethics and institutional requirements.

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
