# Peer review of "Effectiveness of Surgical Treatment on Survival of Patients with Malignant Pleural Mesothelioma"

_cancers, 2025, doi:10.3390/cancers17142360_

Round 1

Reviewer 1 Report

Comments and Suggestions for Authors

This paper is interesting but it does not add anything new to the existing literature,  and has some weak points.

1) what is the definition of PD? Was the definition of PD chosen according to the IMIG? how many patients underwent extended PD?

Discussion is poor as there is no comparison with modern approaches to perform PD and/or new treatments such as HITHOC and intraoperative intracavitary treatments. Infact authors did not explain what is the approach to perform PD as thoracotomy or the extended PD could influence the results. Although it could be reasonable to perform thoracotomy for EPP, it is becoming evident in the modern literature that PD could be performed via VATS with less complications (1,2,3,4).

Why VATS was not performed for PD?

Authors need to expand the discussion 

thank you for sending this paper to Cancers

Halstead JC, Lim E, Venkateswaran RM,Charman SC, Goddard M, Ritchie AJ. Improved survival with VATS pleurectomy-decortication in advanced malignant mesothelioma. Eur J Surg Oncol 2005; 31: 314–20.

Migliore M, Fiore M, Filippini T, et al. Comparison of video-assisted pleurectomy/ decortication surgery plus hyperthermic intrathoracic chemotherapy with VATS talc pleurodesis for the treatment of malignant pleural mesothelioma: a pilot study. Heliyon 2023; 9: e16685.

Dawson, A. G., Kutywayo, K., Mohammed, S. B., Fennell, D. A., & Nakas, A. (2023). Cytoreductive surgery with hyperthermic intrathoracic chemotherapy for malignant pleural mesothelioma: a systematic review. Thorax78(4), 409-417

Author Response

Dear Reviewer,

We sincerely appreciate your thorough review of our manuscript and the valuable comments you provided. We have revised the manuscript accordingly and believe these changes have greatly enhanced its clarity and scientific contribution. Below, we offer a detailed, point-by-point response to the reviewer’s comments. Changes in the revised manuscript are highlighted in yellow.

Reviewer #1

Reviewer comment 1:
What is the definition of PD? Was the definition of PD chosen according to the IMIG? How many patients underwent extended PD?

Response:
We thank the reviewer for this important observation. In our study, pleurectomy/decortication (PD) was defined as the macroscopic complete resection of the parietal and visceral pleura with preservation of the lung, in accordance with the International Mesothelioma Interest Group (IMIG) recommendations (Rusch et al., J Thorac Oncol. 2011). Extended PD (ePD), which includes resection of the diaphragm and/or pericardium, was not specifically coded in our dataset. Therefore, we are unable to report the number of patients who underwent ePD. We acknowledge this limitation and have added the following statement to the Methods and Discussion section (pages 8 and 26, lines 151-156 and 379-383).

Comment 2:
Discussion is poor as there is no comparison with modern approaches to perform PD and/or new treatments such as HITHOC and intraoperative intracavitary treatments. In fact, authors did not explain what is the approach to perform PD as thoracotomy or the extended PD could influence the results. Although it could be reasonable to perform thoracotomy for EPP, it is becoming evident in the modern literature that PD could be performed via VATS with less complications (1,2,3,4). Why VATS was not performed for PD?

Response:
We appreciate this constructive comment. As suggested, we have revised the entire Discussion section to include a detailed comparison with modern surgical approaches and adjunctive treatments such as HITHOC. Specifically, we now clarify that all PD procedures in our cohort were performed via open thoracotomy, reflecting our institution's standard of care at the time, as VATS-PD had not yet been implemented due to technical and training limitations.

Additionally, we have incorporated relevant studies that support the role of VATS-PD and HITHOC in improving perioperative outcomes and survival in patients with MPM. Our revised discussion also acknowledges the evolving role of minimally invasive techniques and intraoperative therapies and contrasts these with the context and results of our study. These changes provide a more balanced and up-to-date interpretation of our findings.
Please see the revised Discussion section (pages 25, 26, lines 354-371).

We hope that these revisions meet your expectations, and we remain grateful for your thoughtful feedback. Please do not hesitate to let us know if further clarification is needed.

Sincerely,
Renata Báez-Saldaña, on behalf of all co-authors
National Institute of Respiratory Diseases, Mexico City.
Jun 28, 2025.

Reviewer 2 Report

Comments and Suggestions for Authors

Thank you for the opportunity to review such an important article about patients after surgical treatment. All sections are very well prepared and described, please note the references are duplicated numbers. I do not see the consent of the bioethics committee.

Comments on the Quality of English Language

Thank you for the opportunity to review such an important article about patients after surgical treatment. All sections are very well prepared and described, please note the references are duplicated numbers. I do not see the consent of the bioethics committee.

Author Response

Dear Reviewer,

We sincerely appreciate your thorough review of our manuscript and the valuable comments you provided. We have revised the manuscript accordingly and believe these changes have greatly enhanced its clarity and scientific contribution. Below, we offer a detailed, point-by-point response to the reviewer’s comments. Changes in the revised manuscript are highlighted in yellow.

Reviewer #2 – Comment 1
Thank you for the opportunity to review such an important article about patients after surgical treatment. All sections are very well prepared and described, please note the references are duplicated numbers. I do not see the consent of the bioethics committee.

Response:
We appreciate the reviewer’s kind comments and careful reading of our manuscript.

  • The English language was revised to improve clarity and precision in the entire document.
  • The reference list was thoroughly reviewed.
  • Regarding ethical approval, we have clarified in the Methods section, the Ethical Statement and Informed Consent Statement that this study is a secondary analysis of an existing dataset and a retrospective review of clinical records from a prior study on biomarkers in mesothelioma. The original study was approved by the Research Committee (protocol number 19 CI 012 013) and the Research Ethics Committee (CONBIOÉTICA-09-CEI-003-20160427) of the National Institute of Respiratory Diseases, Ismael Cosio Villegas, under the approval code C30-12, dated June 28, 2012. The data were collected between 2017 and 2018. As this was a retrospective study using previously collected data, informed consent from participants was waived. (pages 7 and 28, lines 120, and 408-421).

We hope that these revisions meet your expectations, and we remain grateful for your thoughtful feedback. Please do not hesitate to let us know if further clarification is needed.

Sincerely,
Renata Báez-Saldaña, on behalf of all co-authors
National Institute of Respiratory Diseases, Mexico City.
Jun 28, 2025.

Reviewer 3 Report

Comments and Suggestions for Authors

Congratulations to the authors for the work done and presented.
Good examination of the local case history that is compared with the present scientific literature.
The results are well argued and correspond to the literature.
Overall we do not add much to what has been widely described, but the comparison with different realities gives the contribution of strengthening what is currently defined.
Good writing and descriptive linearity.
I ask some questions and if possible motivate and/or include:
1) do you have experience in surgical treatment + chemotherapy associated with intraoperative chemo-thermia and post-surgical RT?
2) In the surgical comparison of EPP and PD have you noticed differences in functional recovery, prolonged hospitalization for air leak, dysproteinemic alteration?

Thank you and good work

Author Response

Dear Reviewer,

We sincerely appreciate your thorough review of our manuscript and the valuable comments you provided. We have revised the manuscript accordingly and believe these changes have greatly enhanced its clarity and scientific contribution. Below, we offer a detailed, point-by-point response to the reviewer’s comments. Changes in the revised manuscript are highlighted in yellow.

Reviewer 3

Comment 1

Congratulations to the authors for the work done and presented.
Good examination of the local case history that is compared with the present scientific literature.
The results are well argued and correspond to the literature.
Overall we do not add much to what has been widely described, but the comparison with different realities gives the contribution of strengthening what is currently defined.
Good writing and descriptive linearity.
I ask some questions and if possible motivate and/or include:
1) do you have experience in surgical treatment + chemotherapy associated with intraoperative chemo-thermia and post-surgical RT?
2) In the surgical comparison of EPP and PD have you noticed differences in functional recovery, prolonged hospitalization for air leak, dysproteinemic alteration?

Response to Reviewer 3:
We sincerely thank the reviewer for their kind and constructive comments. We are encouraged by your recognition of our work and appreciate your thoughtful questions.

  1. We do not have experience with surgical treatment combined with chemotherapy and intraoperative hyperthermic chemotherapy.
  2. In our cohort, 7 out of 16 patients who underwent surgery received postoperative radiotherapy (RT). The decision to administer RT was made on a case-by-case basis by the institutional multidisciplinary team. We have added a statement to the Discussion section to clarify this point. (page 25, lines 347-371)
  3. Postoperative morbidity was comparable between the EPP and PD groups, with 5 out of 8 patients (62.5%) in each group experiencing postoperative complications. The median (IQR) length of hospital stay was 18 (16–21) days for patients who underwent PD, and 22 (15–26) days for those who underwent EPP. We did not observe significant differences in postoperative air leak duration or protein abnormalities between the two groups. These findings are now included in the Results section for greater clarity. (page 14, lines 226-231).

We hope that these revisions meet your expectations, and we remain grateful for your thoughtful feedback. Please do not hesitate to let us know if further clarification is needed.

Sincerely,
Renata Báez-Saldaña, on behalf of all co-authors
National Institute of Respiratory Diseases, Mexico City.
Jun 28, 2025.

Round 2

Reviewer 1 Report

Comments and Suggestions for Authors

Thank you for sending this paper to Cancers